# Rehabilitation Interventions for Post-Acute COVID-19 Syndrome: A Systematic Review

**DOI:** 10.3390/ijerph19095185

**Published:** 2022-04-24

**Authors:** Stefania Fugazzaro, Angela Contri, Otmen Esseroukh, Shaniko Kaleci, Stefania Croci, Marco Massari, Nicola Cosimo Facciolongo, Giulia Besutti, Mauro Iori, Carlo Salvarani, Stefania Costi

**Affiliations:** 1Physical Medicine and Rehabilitation Unit, Azienda Unità Sanitaria Locale-IRCCS di Reggio Emilia, Viale Risorgimento n.80, 42123 Reggio Emilia, Italy; stefania.fugazzaro@ausl.re.it (S.F.); otmen.esseroukh@studenti.unimi.it (O.E.); stefania.costi@unimore.it (S.C.); 2Clinical and Experimental Medicine PhD Program, University of Modena and Reggio, Via del Pozzo n.74, 41100 Modena, Italy; 3Department of Surgical, Medical, Dental and Morphological Sciences Related to Transplant, Oncology and Regenerative Medicine, University of Modena and Reggio Emilia, 41124 Modena, Italy; shaniko.kaleci@unimore.it (S.K.); carlo.salvarani@unimore.it (C.S.); 4Clinical Immunology, Allergy and Advanced Biotechnologies Unit, Azienda Unità Sanitaria Locale—IRCCS di Reggio Emilia, 42123 Reggio Emilia, Italy; stefania.croci@ausl.re.it; 5Infectious Diseases Unit, Azienda Unità Sanitaria Locale—IRCCS di Reggio Emilia, 42123 Reggio Emilia, Italy; marco.massari@ausl.re.it; 6Pulmonology Unit, Azienda Unità Sanitaria Locale—IRCCS di Reggio Emilia, 42123 Reggio Emilia, Italy; nicolacosimo.facciolongo@ausl.re.it; 7Radiology Unit, Azienda Unità Sanitaria Locale—IRCCS di Reggio Emilia, Viale Risorgimento 80, 42123 Reggio Emilia, Italy; giulia.besutti@unimore.it; 8Medical Physics Unit, Azienda Unità Sanitaria Locale—IRCCS di Reggio Emilia, 42123 Reggio Emilia, Italy; mauro.iori@ausl.re.it; 9Rheumatology Unit, Azienda Unità Sanitaria Locale—IRCCS di Reggio Emilia, 42123 Reggio Emilia, Italy

**Keywords:** long COVID, PACS, rehabilitation, post-acute COVID-19 syndrome, pulmonary rehabilitation, exercise

## Abstract

Increasing numbers of individuals suffer from post-acute COVID-19 syndrome (PACS), which manifests with persistent symptoms, the most prevalent being dyspnea, fatigue, and musculoskeletal, cognitive, and/or mental health impairments. This systematic review investigated the effectiveness of rehabilitation interventions for individuals with PACS. We searched the MEDLINE, Embase, Cochrane Register of Controlled Trials, CINHAL, Scopus, Prospero, and PEDro databases and the International Clinical Trials Registry Platform for randomized controlled trials (RCTs) up to November 2021. We screened 516 citations for eligibility, i.e., trials that included individuals with PACS exposed to exercise-based rehabilitation interventions. Five RCTs were included, accounting for 512 participants (aged 49.2–69.4 years, 65% males). Based on the revised Cochrane risk-of-bias tool (RoB 2.0), two RCTs had “low risk of bias”, and three were in the “some concerns” category. Three RCTs compared experimental rehabilitation interventions with no or minimal rehabilitation, while two compared two active rehabilitation interventions. Rehabilitation seemed to improve dyspnea, anxiety, and kinesiophobia. Results on pulmonary function were inconsistent, while improvements were detected in muscle strength, walking capacity, sit-to-stand performance, and quality of life. Pending further studies based on qualitatively sound designs, these first findings seem to advocate for rehabilitation interventions to lessen disability due to PACS.

## 1. Introduction

Long-COVID, also called post-acute COVID-19 syndrome (PACS), is an umbrella term for a complex multisystem secondary condition that follows COVID-19, irrespective of its severity [1]. The terms “long-COVID” and “long-haul COVID” were coined by patients in the first months of the pandemic [2], while the definition of post-acute COVID-19 syndrome was proposed by different authors at the end of 2020 and in 2021 to standardize the patterns of symptoms related to SARS-CoV-2 infection after the acute phase [3,4,5]. Post-acute COVID-19 syndrome is currently defined as a condition characterized by persistent symptoms and/or delayed or long-term complications beyond four weeks from the symptom onset of SARS-CoV-2 infection [4]. PACS can be subdivided into two categories: (1) subacute or ongoing symptomatic COVID-19, including symptoms and abnormalities present from 4–12 weeks beyond acute COVID-19, and (2) chronic or post-COVID-19 syndrome, which includes symptoms and abnormalities persisting or present beyond 12 weeks from the onset of acute COVID-19 [4]. Nalbadian et al. summarized the epidemiology and organ-specific sequelae of post-acute COVID-19 and addressed management considerations for the interdisciplinary comprehensive care of these patients [4]. The five most common symptoms are fatigue (58%), headache (44%), attention disorder (27%), hair loss (25%), and dyspnea (24%) [6], but a variety of other persistent symptoms are reported, including cough, chest pain, myalgia, joint pain, impaired mobility, cognitive impairment (“brain fog”, memory loss), olfactory and gustatory dysfunction, sleep disorders, depression, anxiety, post-traumatic stress disorder, gastrointestinal upset, rashes, and palpitations [5,7,8,9]. Altogether, these symptoms affect the physical, cognitive, and mental functioning of individuals and lead to reduced independence in activities of daily living (ADL) [10] and to an impaired quality of life (QoL) [11]. Taboada et al. reported that, six months after hospital discharge for COVID-19, nearly 50% of patients referred functional limitations in everyday life, focusing attention on the long-term burden of this illness in COVID-19 survivors [10]. A deterioration in QoL was also reported in COVID-19 survivors six months after hospital discharge, with impaired ability to perform activities of daily living (35%), reduced mobility (33%), and pain or discomfort (33%) being the most commonly reported changes [11].

Although the magnitude of this new health condition is still unknown, its prevalence has been estimated to be upwards of 20% of the individuals who have recovered from the acute phase of SARS-CoV-2 infection [12]. Given the millions of individuals worldwide who have been, or will be, affected by COVID-19, the societal impact is likely to be profound and long lasting [12], and an urgent need to investigate the survivorship burden associated with PACS has been advanced [13]. In fact, in contrast with the considerable effort undertaken to understand the acute manifestations of the SARS-CoV-2 infection, the research on the characterization of PACS trajectories and on the impact of specific and comprehensive treatments are ongoing. In this regard, a guideline on the clinical management of COVID-19 patients [14] recognizes that it is a multisystem disease that, in certain cases, may require full multidisciplinary team rehabilitation to enable recovery. Regardless of disease severity, an accurate assessment of physical and cognitive impairments, pain, fatigue, mood disorders, and performance in ADL should be performed in those patients experiencing persistent symptoms and functional limitations, and individualized rehabilitation programs should be suggested accordingly [15]. The training principles of comprehensive pulmonary rehabilitation may be indicated for individuals with persistent fatigue, reduced exercise capacity, and breathlessness, and ADL training or the provision of assistive devices or home accommodations (e.g., mobility aid or toilet grab bars) may be appropriate for a period of time [16]. In patients with chronic conditions, education and advice on helpful strategies should be provided [16]. PACS-related fatigue should be handled through a paced, individualized return to usual activities that is supported by energy conservation techniques, proper diet, hydration, and the management of pain, sleep disorders, and mood disturbances [17]. Coherently, the National Institute for Health and Care Excellence (NICE) guideline recommends integrated multidisciplinary rehabilitation services, which include a range of specialist skills and expertise in managing fatigue and respiratory symptoms for the effective rehabilitation of individuals with long-term effects of COVID-19 [18]. A recent American consensus specifically addressed post-acute persistent breathing discomfort and respiratory sequelae and recommended rehabilitation for people with dyspnea and breathing abnormalities, fatigue, balance impairments, peripheral and pulmonary muscle weakness, and reduced endurance and gait limitation to promote functional improvement and to facilitate a return to activities of daily living [19]. Another consensus suggested a coordinated systematic approach to the evaluation and treatment of patients presenting with post-acute COVID-19-related cognitive symptoms, recommending referral to a specialist (i.e., speech-language pathologist, occupational therapist, neuropsychologist) with expertise in cognitive rehabilitation techniques [20].

However, to date, the evidence on the effectiveness of such programs is limited, and the same guideline calls for research to investigate the effectiveness of rehabilitation interventions and exercise for individuals with PACS.

As such, the primary aim of this systematic review was to explore the effectiveness of rehabilitation interventions for adult patients with PACS by reporting the main changes in outcomes after the experimental interventions. Secondary aims were to describe the characteristics of the rehabilitation interventions currently being investigated in this population and the outcome measures used to verify their effectiveness.

## 2. Materials and Methods

The protocol of this review was registered on PROSPERO, registration number: CRD42022304254.

According to the research question, we included all the trials meeting the following eligibility criteria:

Participants: Adults (≥18 years) diagnosed with COVID-19 at least four weeks before study enrollment, according to the definition of PACS [4]. 

Interventions: Rehabilitation that included any type of exercise, i.e., outpatient or home-based interventions based on aerobic and/or resistance exercise, respiratory physiotherapy, relaxation techniques, yoga, or other interventions, including stretching, also when associated with other multimodal programs, e.g., cognitive or psychosocial interventions. Studies including drugs and/or dietary supplements associated with exercise were considered eligible. 

Comparison: Usual care or any other comparator intervention.

Outcomes: Any outcome measure. 

Study design: Randomized controlled trials (RCTs)

We excluded trials conducted on hospitalized COVID-19 patients, trials whose experimental intervention was limited to drugs or dietary supplements and that did not include exercise, and non-randomized study designs, observational studies, case series, and case reports. 

### 2.1. Information Source and Search Strategy

An electronic search was conducted in October and November 2021. The following databases were consulted up to 4 November 2021: MEDLINE, Embase, Cochrane Register of Controlled Trials, CINAHL, Scopus, Prospero, and PEDro. No restrictions were placed on the publication date. Study reports in English, Italian, Spanish, and French were accepted. The search strategy with keywords is reported in Appendix A. The International Clinical Trials Registry Platform (ICTRP) (WHO) was also consulted to search for study protocols matching the inclusion criteria. The corresponding authors were contacted to ask for unpublished data or pre-prints, if available. 

A hand search of references in relevant articles was performed for further literature. 

### 2.2. Data Collection and Analysis 

Three researchers (AC, OE, and SC) independently screened the results of the electronic search for eligibility by screening titles and abstracts. In cases of disagreement, the authors screened the full-text publication, when available [21]. SF checked ICTRP for protocols and contacted 13 corresponding authors of the research protocols that were eligible, asking for data that could contribute to this review. Three replied to the request, declaring that the studies were ongoing or that the data were not yet available for analysis. 

Figure 1 presents the study selection process in a flow diagram, as recommended in the PRISMA statement [22], which shows the total number of retrieved references and the number of included and excluded studies. 

### 2.3. Data Extraction and Management 

Two researchers (AC and OE) independently extracted the following data: (1) general information, such as authors, publication date, and country; (2) study characteristics, such as the number of participants, inclusion and exclusion criteria, timing since COVID-19 diagnosis, and dropouts; (3) characteristics of experimental interventions, such as type, duration, and intensity of exercise, setting, and modality (supervised/unsupervised); (4) characteristics of standard care/control; (5) duration of follow-up; (6) any outcome measure and results for before-after and between-group comparisons, using the risk ratio for binary outcomes and mean difference for continuous outcomes; (7) adverse events reported by authors. Data were extracted from the original reports or through correspondence with the authors. When this was not possible, i.e., after a minimum of three attempts to obtain data from the corresponding authors, data were estimated by the researchers based on the available means and standard deviations. Missing *p*-values of the mean differences of between-group comparisons were estimated using the Student’s *t*-test, assuming a similar variability of distributions, using STATA version 17. Missing standard deviations were estimated, assuming that the mean distributions had equal standard deviations. 

Given the complexity of the interventions being investigated, to decide which trials were eligible for this review, the researchers tabulated the main characteristics of the intervention and compared them with all the eligible interventions. Disagreements were discussed with SF and SC and resolved.

Two researchers (AC and OE) independently assessed the risk of bias of each trial using the Cochrane risk-of-bias 2 tool for RCTs [23], classifying them as either at “high risk”, “some concerns”, or “low risk” of bias. Disagreements were discussed with SC and resolved. In the cases of risk of bias due to missing results, the researchers contacted the corresponding authors to obtain relevant information. 

### 2.4. Synthesis

The review findings are presented descriptively. To achieve complete and transparent reporting and to facilitate interpretability, the eligible trials are summarized in tables that describe the general characteristics of each study design and the results obtained by outcome domain. Moreover, the characteristics of the experimental interventions and controls and the outcome measures used in each study design are described in detail in Appendix A. The trials are ordered from low to high risk of bias to increase the prominence of the most trustworthy evidence.

## 3. Results

### 3.1. Study Selection 

The results of several searches that contributed to this review are outlined in the PRISMA flow diagram depicted in Figure 1.

A total of 516 citations were screened for inclusion; of these, 419 were retrieved from electronic database searches, 50 were retrieved from the databases of ongoing studies, and 47 were retrieved from cross-referencing. After excluding citations based on the screening of titles and abstracts, 15 were deemed potentially relevant for further evaluation. Fourteen full texts and one abstract with sufficient information were assessed for eligibility; of these, five trials met the inclusion criteria and were included in this review [24,25,26,27,28]. Data were extracted from all the study reports and the Appendix A wherever possible [24,25], and the corresponding authors were contacted on multiple occasions to retrieve missing information [24,26,27,28].

### 3.2. Methodological Quality 

Figure 2 shows the results of the risk-of-bias assessment. Underreporting hindered the assessment of methodological quality, particularly for the trial by De Souza and collaborators [26], which was reported only as an abstract.

Of the five RCTs included, two were identified as having a “low risk of bias” [24,25], while the other three were in the “some concerns” category [26,27,28]. 

All of the trials reported having used random assignment, by means of different strategies, but specific procedures for generating the allocation sequences were described in detail in two reports only [24,25]. 

As usual for trials in the rehabilitation field, the blinding of participants and professionals who implemented the interventions was not possible. However, in all but one [26] of the RCTs included in this review, the objective functional outcomes were collected by blind assessors, whereas dyspnea, fatigue, and quality of life were self-reported.

Attrition at the final follow-up ranged from 0% [27] to 32.1% [26], with a mean final attrition of 12% of baseline enrollment (Table 1).

One RCT analyzed the data of participants using the intention-to-treat approach and imputed data for dropouts [24], while the other four RCTs [25,26,27,28] did not impute data for dropouts, making their results vulnerable to attrition bias. The statistical analysis was not sufficiently described in one study [26].

### 3.3. Description of Included Trials

The characteristics of the included trials are described in Table 1.

The five RCTs included in this review investigated 512 participants (65% males), aged 49.2 to 69.4 years, who suffered from persistent symptoms of PACS. Due to cultural restrictions regarding the participation of females in research studies, one study [25] included only male participants.

Patients were assessed at the baseline and at the short-term follow-up, which took place immediately after the completion of the intervention in all the RCTs included in this review. Some trials provided for multiple follow-ups over a period ranging from four to 28 weeks from the baseline [24,25].

#### 3.3.1. Interventions

Appendix A shows the characteristics of the experimental interventions and controls. Three of the five included trials [24,26,27] assessed the effect of the experimental intervention in comparison to no rehabilitation/drug-only intervention [26,27] or minimal rehabilitation intervention consisting of a short educational instruction program provided at the baseline that also included the recommendation to take part in moderate physical activity [24].

Two trials compared two active interventions. Nambi and collaborators compared two programs of supervised aerobic and resistance training, which were different in terms of the intensity of the aerobic training [25]. Srinivasan and collaborators compared a respiratory physiotherapy program based on a combination of pursed lip breathing exercises and bhastrika pranayama (a yoga and pranayama form of breathing exercise), with the breathing exercises performed with incentive spirometry [28].

In the fifth study, the intervention consisted of a six-week low-intensity pulmonary rehabilitation program, with no further details provided [26].

Exposure ranged from 120 min [27] to 48 h [25], delivered over the course of six [24,26,27,28] to eight weeks [25]. Median exposure was approximately 2¼ h/week. Intervention settings varied from home-based [24,26,28] to the outpatient physiotherapy department [25], and the delivery modality was equally distributed between unsupervised [24,28] and supervised [25,26].

Only Li and collaborators recorded the adverse events associated with the interventions and reported them in detail in a supplementary table [24]. Overall, 174 adverse events occurred, most of them classified as very mild to moderately severe and ranging from chest tightness to feelings of weakness or reduced physical strength and cough. No serious adverse event occurred during the study period, while during the follow-up, eight individuals (7.62%), five in the experimental group and three in the control group, were hospitalized for non-life-threatening events. None of those events were associated with COVID-19 or to the experimental intervention.

#### 3.3.2. Outcome Measures

As shown in Appendix A, the RCTs included in this review assessed the domains of symptoms, functional outcomes and quality of life (QoL), and independence in patients with PACS using 22 different outcome measures. Three study designs measured all these domains [24,25,27].

Dyspnea and fatigue were measured using the modified British Medical Research Council Dyspnea Score (mMRC) and the 10-point version of the Borg Rating of Perceived Exertion (RPE) [24,26]. Mood disturbances were measured by the Zung Self-Rating Anxiety Scale (SAS) [27], the Zung Self-Rating Depression Scale (SDS) [27], and the Tampa Scale for Kinesiophobia (TSK-11) [25].

Functional outcomes included pulmonary function tests, muscle mass and strength, and functional exercise capacity. Pulmonary function was measured by spirometry to record the static and dynamic lung volumes [24,27,28], and in a few study designs, also by the diffusing lung capacity for carbon monoxide (DLCO%) [27], the maximum voluntary ventilation (MVV), and the peak expiratory flow (PEF) [24]. Muscle mass was measured by cross-sectional magnetic resonance imaging (MRI) of the major appendicular muscles [25], while muscle strength was measured by means of both the handgrip strength test [25] and the static squat test at the wall [24]. Functional exercise capacity was measured by the 6-min walking test (6MWT) [24,27] or by the 30 Second Sit to Stand Test (STS) in addition to the physical activity in daily life (PADL) [26].

Quality of life was measured using the 12-item or the 36-item versions of the Short Form Health Survey [24,27] or the Sarcopenia and Quality of Life questionnaire (SarQoL) [25]. Independence was measured by only one trial [27], by means of the Functional Independence Measure (FIM).

Table 2, Table 3 and Table 4 present the results of the within- and between-group comparisons for all the outcomes assessed in the trials included in this review.

#### 3.3.3. Symptoms

Two studies assessed the effect of rehabilitation interventions on dyspnea and fatigue in patients with PACS [24,26]. Both these trials compared comprehensive rehabilitation to no intervention [26] or to short educational instruction at baseline [24], and both detected an improvement in dyspnea and perceived exertion in favor of the experimental group at the end of the intervention (six weeks). However, this gain was not maintained at either the medium- or long-term follow-ups [24].

Regarding mood disturbances, a significant reduction in the anxiety rate was detected in one trial [27] following a six-week respiratory physiotherapy program, compared to no rehabilitation.

The fear of movement was measured in one trial that compared low-intensity versus high-intensity aerobic training; a significant improvement in kinesiophobia was detected in favor of the low-intensity group at all the follow-ups [25].

#### 3.3.4. Functional Outcomes

Three studies assessed the effect of a rehabilitation intervention on the pulmonary function of patients with PACS [24,27,28]. While one trial did not detect any effect of rehabilitation [24], the second recorded a significant improvement in all the parameters investigated following respiratory muscle training and home exercise, compared to no rehabilitation [27]. The third trial found a significant improvement in FEV1 following breathing exercises with bhastrika pranayama, compared to incentive spirometry [28].

One trial investigated the muscle mass of the major appendicular muscles and found that it improved equally following both high- and low-intensity aerobic training [25]. Conversely, the same trial showed a significant improvement in handgrip strength only in the high-intensity training group, a result confirmed by Li and collaborators, who found an improvement in the strength of lower limbs following home-based exercise [24].

Finally, improvements in functional exercise capacity following experimental rehabilitation were detected through the distance walked in six minutes in two trials [24,27] and through sit-to-stands performed in 30 s and the steps walked in a day in a third one [26].

#### 3.3.5. Quality of Life and Independence Outcomes

Three trials included in this review [24,25,27] reported at least one health-related QoL outcome. Significant improvements in favor of experimental rehabilitation were detected by one trial on the SarQoL; this gain was confirmed at all the assessment times (four weeks, eight weeks, and six months after the baseline) [25]. Short-term benefits of rehabilitation were also detected in all the domains of the SF-36 [27] and in the physical component of the SF-12 [24].

Independence was measured by only one trial, which detected no significant differences in the FIM between the level of assistance needed by patients who performed respiratory physiotherapy and that of those who received no intervention [27].

## 4. Discussion

The aim of this review was to explore the effectiveness of experimental rehabilitation interventions for patients with PACS and to describe the characteristics of those interventions and the outcome measures used to verify their effectiveness.

### 4.1. Main Findings

Rehabilitation seemed to improve dyspnea, anxiety, and kinesiophobia. Results on pulmonary function were inconsistent, while improvements were detected in muscle strength, walking capacity, sit-to-stand performance, and quality of life. The main findings of this systematic review require some reflection.

The studies included in this systematic review focused on patients enrolled in trials at least four weeks after COVID-19 diagnosis. For most of the patients, this timing corresponded to the post-acute phase of the disease, that is, between four and twelve weeks after acute infection [4]. Differently from others, Liu and collaborators enrolled patients six months after the onset of the disease, i.e., the ‘chronic/post-COVID-19 condition’ phase [4].

Researchers of the five trials included in this review used a variety of outcome measures (22 in all) to assess the health domains attributable to physical symptoms, psychological impairments, limitations in function, QoL, and independence. This is in line with the recommendation made early after the onset of the pandemic when, in the absence of evidence, experts reached a consensus on the need for a comprehensive assessment of rehabilitation needs in post-acute COVID-19 patients [16]. 

Assessments of pulmonary function, exercise capacity, and QoL were recurrent in the RCTs included in this systematic review. However, the presence of missing data and the high heterogeneity of the outcome measures used prevented us from carrying out a meta-analysis. 

Three out of the five trials [24,25,26] investigated the effectiveness of comprehensive rehabilitation interventions, which consisted of respiratory physiotherapy associated with aerobic and resistance training [24,25]; these may, to some extent, meet the requirements for pulmonary rehabilitation. We could not retrieve detailed information on the components of the intervention applied by De Souza and collaborators [26], but as this was defined in the report as a “pulmonary rehabilitation”, we assume that it met the requirements for this definition [29].

The experimental rehabilitation interventions, all delivered in a time span of 6–8 weeks, were all quite heterogeneous in the intensity and frequency of activities, as some protocols contemplated exercising several times a day, while others several times a week (see Appendix A for more details). The most demanding schedule consisted of six hours per week. 

Half of the interventions were supervised in person or through videoconference. The others were based on self-managed programs, as also recommended by the WHO in its guideline to support self-management for individuals recovering from COVID-19 [14]. 

Regarding the methodological quality of the RCTs included, overall, it was sufficient, as two trials were judged as having “low risk” of bias and the other three in the “some concerns” category of the RoB 2 tool. 

Considering that the RCTs were conducted during a pandemic that put health organizations under extreme pressure, it is plausible that it was not always possible to limit all the potential risks of bias.

### 4.2. Theoretical and Practical Implications of This Study

All the RCTs included in this review demonstrated the effectiveness of the experimental exercise-based intervention on some of the outcomes assessed. 

We think that the choice to include both aerobic and resistance exercise in the experimental rehabilitation interventions targeted to this population was based on the need to address some of the most frequent persistent symptoms after COVID-19, i.e., dyspnea, fatigue, and sarcopenia. Indeed, patients with PACS also manifest impaired psychological status and exercise capacity, activity limitations, and a worsened QoL, which benefit from pulmonary rehabilitation when applied to chronic respiratory conditions [29]. As a matter of fact, the study that assessed the fear of movement and its association with thoughts and beliefs about pain [25] found a significant improvement in an intervention group practicing low-intensity aerobic exercise at the end of the four-week intervention. Furthermore, this improvement was maintained at both the medium- (eight weeks) and at the long-term (six months) follow-ups, in line with the data from previous pulmonary rehabilitation literature [30]. Liu and collaborators found a significant reduction in anxiety in the experimental group [27]. This is perhaps due to the fact that depressive disorders require a specially designed exercise program that includes motivational support [31]. Two studies [24,26] analyzed the effect of a rehabilitation program on dyspnea and detected a significant improvement immediately after; in one case, however, those results were not maintained after 28 weeks of the follow-up [24].

Two of the five RCTs included in this review investigated the effectiveness of rehabilitation approaches that were chiefly based on respiratory physiotherapy techniques, either associated with a yoga technique [28] or with stretching [27]. They registered significant gains in pulmonary function [27,28] and functional exercise capacity [27]. Pulmonary rehabilitation consists of a multidisciplinary approach based on exercise training [32] and is designed to improve the physical and psychological condition of individuals with chronic respiratory diseases. Its implementation has been advocated since the very first months of the pandemic, to be performed in a safe manner if patients were still contagious [33]. Li and collaborators demonstrated the benefits of an unsupervised six-week home-based pulmonary rehabilitation program delivered via smartphone and remotely monitored with heart rate telemetry [24]. Long-term positive effects were detected in exercise capacity and muscle strength. This trial failed to demonstrate significant differences in lung function parameters after rehabilitation. On the other hand, three other RCTs detected significant improvements in some lung function tests. A benefit in muscle strength following rehabilitation was also highlighted by Nambi and collaborators [25], although this was not associated with any change in muscle mass. Of note, in this trial, two active rehabilitation interventions were compared, which differed only in terms of the intensity of the 30 minutes of aerobic training (60–80% max HR, versus 40–60% max HR).

Moreover, functional exercise capacity improved significantly in patients exposed to experimental interventions compared to the controls in all the trials that assessed this specific domain, irrespective of the outcome measure used [24,26,27]. 

As for QoL and independence, two trials [25,27] found a beneficial effect of experimental rehabilitation interventions on QoL of individuals with PACS, which is consistent with improvements in QoL described after pulmonary rehabilitation in patients with COPD [34], those with asthma [35], and those who have undergone lung cancer surgery [36]. 

The only study exploring the effect of respiratory rehabilitation on FIM found no significant improvement. However, this may be due to the good level of independence in B-ADL at the first evaluation (mean score = 109) and to a ceiling effect of the scale.

These results are in line with the current literature, although no research group has yet explored the effectiveness of the intervention using a systematic review approach. Improvements in exercise capacity, HRQoL, dyspnea, fatigue, anxiety, and depression after a pulmonary rehabilitation program were also reported by Soril et al. in a recent rapid review of the literature, including only one RCT [27] and other experimental or quasi-experimental studies on COVID-19 survivors in the first three months after hospital discharge [37].

### 4.3. Limitations

This review has some limitations. As already stated, the outcome measures applied in the RCTs included were highly heterogeneous. Moreover, some of the reports lacked complete data. Both these limitations prevented us from conducting a meta-analysis.

Furthermore, despite the comprehensive search strategy adopted, it is possible that we did not identify all the existing reports of trials that would have been eligible for inclusion, for instance, reports in original languages other than those known by the research team. Furthermore, although we contacted all the corresponding authors of the reports with missing data and sent them follow-up emails if they did not answer, not all the missing data could be retrieved from the corresponding authors. Thus, a complete description of the study procedures, interventions, and results was not possible.

As a result of our search, we found a paucity of trials investigating the effectiveness of rehabilitation in PACS; the vast majority of the studies retrieved focused on the acute phase of COVID-19.

Overall, the results of this review encourage the implementation of rehabilitation in patients with PACS, as its effectiveness seems to be demonstrated, although not always consistently, in all the domains investigated by the trials included in this review. However, as only five RCTs were included, and some of them involved a small number of participants [25,26,27] or raised some concern regarding their internal validity [26,27,28], it is possible that future research will come to different conclusions.

Moreover, as individuals with major post-COVID sequelae such as cerebrovascular disease were frequently not enrolled in the trials included in this review, the generalizability of our results to a wider PACS population is not guaranteed. Furthermore, the average age of the participants in the included studies was relatively high, limiting the information available on the effectiveness of rehabilitation in younger PACS individuals. Finally, only one trial recorded the adverse events associated with experimental rehabilitation [24], in spite of the relevance of this information to clinicians, who, in the absence of strong evidence, must balance elements for and against when indicating what rehabilitation is to be prescribed.

However, this review also has some strengths. In light of the increasing burden of PACS all over the world, this, as far as we know, is the very first systematic review that shows the effectiveness of rehabilitation interventions in the post-acute phase of COVID-19. Furthermore, it provides an overview of the rehabilitation interventions that have been experimented and the outcome measures used to verify their effectiveness.

### 4.4. Future Directions

Larger and adequately powered studies are required to confirm these initial findings. Of note, as post-acute COVID-19 syndrome is expected to represent a health problem for the foreseeable future, it is likely that the literature in this field will rapidly expand.

Based on this information, we can suggest that future studies will clarify the rationale and describe in detail both the experimental interventions in all their components and the controls applied. Moreover, we highlight the need for a common assessment strategy of post-COVID sequelae in order to compare the results of the different trials. A post-COVID Core Outcome Set (PC-COS) group is working on behalf of the World Health Organization (WHO) to create a core set of outcomes to be used in all research studies and in the clinical care of individuals with post-COVID conditions [32].

In addition, we advise researchers in this field to record the feasibility and safety of the interventions and to measure their effectiveness in the medium and long term. In fact, as this is a new post-acute condition with a spontaneous positive trend, it is of the utmost importance to collect data that enable clinicians to weigh the cost-effectiveness of such approaches and to identify those patients that might benefit most from them. Unfortunately, to our knowledge, there are no worldwide disease registries for this syndrome, which makes the tracking and monitoring of affected individuals virtually impossible. The introduction of a PACS disease registry would be extremely useful in the management of its associated physical, psychological, and social sequelae. Moreover, given the sheer size of the population that could potentially suffer from PACS in the near future, research investments are likely to be necessary to support further studies in this field.

## 5. Conclusions

Although recommendations for the rehabilitation of the post-COVID condition already exist [14,19,20,38,39], there is still a lack of evidence on the effectiveness of these interventions in individuals with PACS. This systematic review helped to fill this knowledge gap and suggest that rehabilitation interventions may be effective in addressing the sequelae of COVID-19.

Certainly, the pandemic continues to require great economic and organizational effort of healthcare systems, and carrying out a valid RCT in these times is not within everyone’s reach. Notwithstanding this, given that the number of individuals with long-term consequences of COVID-19 continues to grow, future research should also answer the WHO call for standardized assessment and appropriate treatment. This review can contribute to the planning of sound study designs and the clinical data collection in this population, as it also describes the rehabilitation interventions that have been experimented and reports all the outcome measures used to verify their effectiveness.

## Figures and Tables

**Figure 1 ijerph-19-05185-f001:**
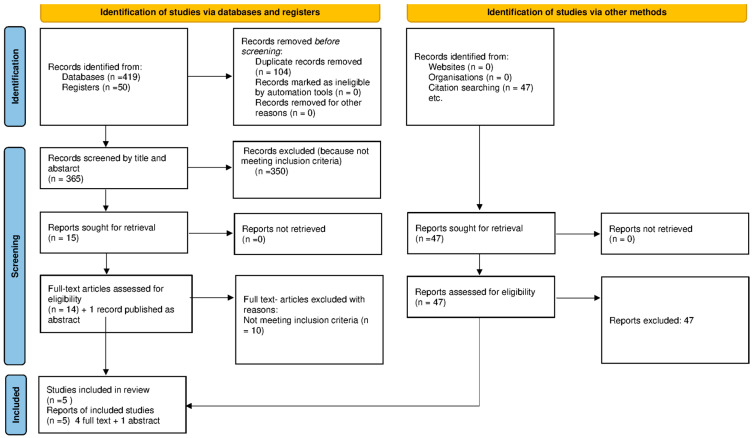
PRISMA 2020 flow diagram for new systematic reviews that includes searches of databases, registers, and other sources.

**Figure 2 ijerph-19-05185-f002:**
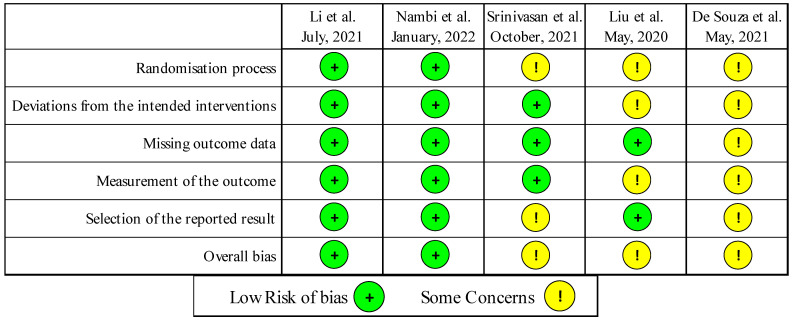
Risk of bias summary: review authors’ judgements about each risk of bias item for each included study.

**Table 1 ijerph-19-05185-t001:** Characteristics of included studies.

Study	Country	N° of Participants			Inclusion Criteria	Exclusion Criteria	Follow Up	Dropout N°/Rate			
		Experi-mental Group	Control Group	Total				Time Points	Experimental Group	Control Group	Total
**Li et al., 2021**	**China**	59	61	120	- Formerly hospitalized COVID-19 survivors - mMRC dyspnea 2–3	- Resting heart rate > 100 bpm - Uncontrolled hypertension - Uncontrolled chronic disease - Cerebrovascular disease within 6 months - Intra-articular drug injection or surgical treatment of lower extremities within 6 months - Use of medication affecting cardiopulmonary function - Inability to walk independently with assistive device - Inability or Unwillingness to collaborate with assessments - Enrollment or participation in other trials within past 3 months - History of severe cognitive or mental disorder or substance abuse - Enrollment in other rehabilitation program	6 weeks (post treatment) 28 weeks (follow up)	6 weeks	7 (11.9%)	1 (1.6%)	8 (6.7%)
								28 weeks	2 (3.8%)	5 (8.3%)	7 (6.3%)
								Total	9 (15.3%)	6 (9.8%)	15 (12.5%)
**Nambi et al.,** **2022**	**Saudi Arabia** **Egypt**	38	38	76	- Men aged 60–80 - Post COVID-19 sarcopenia (appendicular skeletal muscle mass index score <7.0 kg/m^2^) - Normal VO2 max (17–18 mL/kg/min) - Normal resting heartbeat (70–90 beats/min)	- Low muscle mass in observation - Handgrip strength less than 24 kg - Slow gait speed (<0.7 m/s) - Prior exercise training, under medication, history of lower limb surgeries, fractures, cardiac problems, respiratory problems, neurological problems, systemic problems, and any other contraindications for aerobic training	4 weeks 8 weeks 6 mo	4 weeks	0	0	0
								8 weeks	1 (2.6%)	2 (5.3%)	3 (3.9%)
								6 m	3 (8.1%)	1 (2.8%)	4 (5.5%)
								Total	4 (10.5%)	3 (7.9%)	7 (9.2%)
**Srinivasan et al., 2021**	**India**	24	24	48	- Patients at post COVID-19 follow up clinic - Aged 18–60	- Post COVID-19 cerebrovascular accident - Post COVID-19 renal failure - Post COVID-19 myocardial infarction	6 weeks	6 weeks	0	0	0
**Liu et al., 2020**	**China**	36	36	72	- Patients with a definite diagnosis of COVID-19 - Age ≥ 65 y - ≥ 6 mo after the onset of other acute diseases - MMSE score > 21 - No COPD or any other respiratory disease - FEV1 ≥ 70%	- Moderate or severe heart disease (Grade III or IV NYHA) - Severe ischemic or hemorrhagic stroke - Severe neurodegenerative diseases	6 weeks	6 weeks	2 (5.5%)	2 (5.5%)	4 (5.6%)
**De Souza et al., 2021**	**Brazil**	104	92	196	- Post COVID-19 phase - Not requiring ICU admission		6 weeks	6 weeks	16 (15.4%)	47 (51.1%)	63 (32.1%)

*mMRC*, modified Medical Research Council; *mo*, months; *y*, *years*; *NYHA*, New York Heart Association.

**Table 2 ijerph-19-05185-t002:** Symptom-related outcomes.

SYMPTOMS										
**Fatigue and Dyspnea Sensation**										
**mMRC:** The modified British Medical Research Council Dyspnea Scale for perceived dyspnea, to favorable outcome.										
**Li et al** **., 2021**	**Pre**	**Post**	**Follow** **-Up**		**Mean Difference for Before-After Comparison**			**Mean Difference (CI) for Between-Group Comparison**		
					**Pre-Post**	**Pre-Follow Up**	***p* value**	**Post Intervention**	**Follow** **-Up**	
Intervention *n =* 59	2	0 (0–0)	0 (0–1)		90.4	72.0	<0.001 *	1.46 (1.17 to 1.82) *p* value < 0.001	1.22 (0.92 to 1.61) *p* Not significant	
Control *n* = 61	2	0 (0–1)	0 (0–1)		61.7	60.0	<0.001 *			
**RPE:** The 10-point Borg Scale, used to measure the level of physical strain or perceived exertion.										
**De Souza et al** **., 2021**	**Pre**		**Post**		**Mean Difference for Before-After Comparison**			**Mean Difference (CI) for Between-Group Comparison**		
					***p* value**			**Post**	***p* value**	
Intervention *n* = 104	4.5 ± 2.6		1.1 ± 1.6		<0.05			−3.1 (−4.2 to −1.5)	<0.001	
Control *n* = 92	4.6 ± 2.2		4.3 ± 2.3		Not significant					
**Mood Disturbances**										
**SAS:** Zung Self Rating Anxiety Rate, a self-report assessment tool that measures levels of anxiety in patients who have anxiety-related symptoms.										
**Liu et al** **., 2020**	**Pre**		**Post**		**Mean Difference for Before-After Comparison**			**Mean Difference (CI) for Between-Group Comparison**		
					***p* value**			**Post Test**	***p* value**	
Intervention *n* = 36	56.3 ± 8.1		47.4 ± 6.3		*p* < 0.05			−4.66 (−10.71 to −4.29) *	*p* < 0.05	
Control *n* = 36	55.8 ± 7.4		54.9 ± 7.3		Not significant					
**SDS:** Self Rating Depression Rate, a short self-rated scale that assesses the psychological and somatic symptoms of depression.										
**Liu et al** **., 2020**	**Pre**		**Post**		**Mean Difference for Before-After Comparison**			**Mean Difference (CI) for Between-Group Comparison**		
					***p* value**			**Post Test**	***p* value**	
Intervention *n* = 36	56.4 ± 7.9		54.5 ± 5.9		Not significant			−0.84 (−4.37 to 1.77) *	Not significant	
Control *n* = 36	55.9 ± 7.3		55.8 ± 7.1		Not significant					
**TSK-11:** Tampa Scale for Kinesiophobia.										
**Nambi et al., 2022**	**Pre**	**Post**	**Follow-Up 1**	**Follow-Up 2**	**Mean Difference for Before-After Comparison**			**Mean Difference for Between-Group Comparison**		
					**Pre-Post**	**Pre-Follow-Up 1**	**Pre-Follow-Up 2**	**Post Intervention**	**Follow Up 1**	**Follow Up 2**
Intervention *n* = 38	32.3 ± 0.9	29.9 ± 0.9	24.5 ± 1.4	18.2 ± 1.0	*p* < 0.001	*p* < 0.001	*p* < 0.001	31.0 (5.99 to 6.81) * *p* value < 0.001 *	24.08 (5.96 to 7.04) * *p* value <0.001 *	20.49 (4.24 to 5.16) * *p* value <0.001 *
Control *n =* 38	32.1 ± 1.0	23.5 ± 0.9	18.0 ± 0.9	13.5 ± 1.0	*p* < 0.001	*p* < 0.001	*p* < 0.001			

Data were extracted by the original reports, or through correspondence with authors, or they were calculated based on available data (*), if possible, when it was not possible to obtain them from the original authors. In these results, *n* refers to the number of participants included in the analyses and is not necessarily equivalent to the number enrolled at the baseline or retained at the follow-up. *CI*, confidence interval.

**Table 3 ijerph-19-05185-t003:** Functional outcomes.

FUNCTIONAL OUTCOMES												
**Pulmonary Function**												
**FEV1 (L):** Forced expiratory volume in the first second is the amount of air you can force from your lungs in one second.												
**Li et al., 2021**	**Pre**	**Post**	**Follow-Up**		**Mean Difference ±SD for Before-After Comparison**			**Mean Difference (CI) for Between-Group Comparison**				
					**Pre-Post**	**Pre-Follow Up**	***p* value**	**Post Intervention**		***p* value**	**Follow-Up**	***p* value**
Intervention *n =* 59	2.24 ± 0.74	2.47 ± 0.65	2.43 ± 0.55		0.28 ± 0.51	0.29 ± 0.48	Not significant *	0.08 (−0.08 to 0.25)		Not significant	0.00 (−0.18 to 0.17)	Not significant
Control *n* = 61	2.14 ± 0.69	2.37 ± 0.77	2.48 ± 0.72		0.18 ± 0.53	0.29 ± 0.43	Not significant *					
**Srinivasan et al., 2021**	**Pre**		**Post**		**Mean Difference for Before-After Comparison**			**Mean Difference (CI) for Between-Group Comparison**				
					**Pre-Post**		***p* value**	**Post Test**			***p* value**	
Intervention *n* = 24	60.04 ± 5.61		75.75 ± 3.80		10.58		<0.0001	5.28 (12.03 to 5.39) *			<0.0001	
Control *n* = 24	63.58 ± 7.25		67.04 ± 7.14		6.40		<0.0001					
**Liu et al., 2020**	**Pre**		**Post**		**Mean Difference for Before-After Comparison**			**Mean Difference (CI) for Between-Group Comparison**				
					***p* value**			**Post Test**			***p* value**	
Intervention *n* = 36	1.10 ± 0.08		1.44 ± 0.25		*p* < 0.05			2.66 (0.05 to 0.32) *			*p* < 0.05	
Control *n* = 36	1.13 ± 0.14		1.26 ± 0.32		Not significant							
**FVC (L):** Forced vital capacity is the total amount of air you can forcibly exhale after the deepest inhalation possible.												
**Li et al., 2021**	**Pre**	**Post**	**Follow-Up**		**Mean Difference ± SD for Before-After Comparison**			**Mean Difference (CI) for Between-Group Comparison**				
					**Pre-Post**	**Pre-Follow-Up**	***p* value**	**Post Intervention**		***p* value**	**Follow-Up**	***p* value**
Intervention *n =* 59	2.85 ± 0.75	2.97 ± 0.75	3.00 ± 0.60		0.21 ± 0.47	0.30 ± 0.38	Not significant *	0.02 (−0.14 to 0.18)		Not significant	0.01 (−0.16 to 0.17)	Not significant
Control *n* = 61	2.69 ± 0.87	2.93 ± 0.91	3.04 ± 0.85		0.19 ± 0.40	0.27 ± 0.43	Not significant *					
**Srinivasan et al., 2021**	**Pre**		**Post**		**Mean Difference for Before-After Comparison**			**Mean Difference (CI) for Between-Group Comparison**				
					**Pre-Post**		***p* value**	**Post Test**			***p* value**	
Intervention *n* = 24	65.88 ± 5.19		70.50 ± 5.53		5.29		<0.0001	0.66 (−2.21 to 4.38) *			Not significant	
Control *n* = 24	67.04 ± 5.18		69.42 ± 5.81		5.35		<0.0001					
**Liu et al., 2020**	**Pre**		**Post**		**Mean Difference for Before-After Comparison**			**Mean Difference (CI) for Between-Group Comparison**				
					***p* value**			**Post Test**			***p* value**	
Intervention *n* = 36	1.79 ± 0.53		2.36 ± 0.49		*p* < 0.05			2.73 (0.08 to 0.48) *			*p* < 0.05	
Control *n* = 36	1.77 ± 0.64		2.08 ± 0.37		Not significant							
**FEV1/FVC**												
**Li et al., 2021**	**Pre**	**Post**	**Follow Up**		**Mean Difference ± SD for Before-After Comparison**			**Mean Difference (CI) for Between-Group Comparison**				
					**Pre-Post**		**Pre-Follow-Up**	***p* value**	**Post Intervention**	***p* value**	**Follow-Up**	***p* value**
Intervention *n =* 59	0.79 ± 0.14	0.84 ± 0.09	0.81 ± 0.09		0.04 ± 0.17		0.02 ± 0.18	Not significant *	0.03 (−0.02 to 0.07)	Not significant	−0.01 (−0.05 to 0.03)	Not significant
Control *n* = 61	0.81 ± 0.12	0.81 ± 0.11	0.82 ± 0.09		0.01 ± 0.16		0.02 ± 0.15	Not significant *				
**Liu et al., 2020**	**Pre**		**Post**		**Mean Difference for Before-After Comparison**			**Mean Difference (CI) for Between-Group Comparison**				
					***p* value**			**Post Test**			***p* value**	
Intervention *n* = 36	60.48 ± 6.39		68.19 ± 6.05		*p* < 0.05			4.73 (4.03 to 9.89) *			*p* < 0.05	
Control *n* = 36	60.44 ± 5.77		61.23 ± 6.43		Not significant							
**MVV (L/min):** Maximal voluntary ventilation is the maximum amount of air that can be breathed in and blown out over a sustained interval, such as 15 or 20 s.												
**Li et al., 2021**	**Pre**	**Post**	**Follow Up**		**Mean Difference ± SD for Before-After Comparison**			**Mean Difference (CI) for Between-Group Comparison**				
					**Pre-Post**		**Pre-Follow Up**	***p* value**	**Post Intervention**	***p* value**	**Follow-Up**	***p* value**
Intervention *n =* 59	74.30 ± 30.60	86.82 ± 28.51	89.17 ± 27.06		14.49 ± 21.60		18.47 ± 22.31	Not significant *	10.57 (3.26 to 17.88)	<0.05	5.20 (−2.33 to 12.73)	Not significant
Control *n* = 61	63.05 ± 26.12	70.87 ± 30.70	80.65 ± 35.96		5.61 ± 17.31		13.81 ± 20.78	0.014 *				
**PEF (L/s):** Peak expiratory flow is a person’s maximum speed of expiration, as measured with a peak flow meter.												
**Li et al., 2021**	**Pre**	**Post**	**Follow-Up**		**Mean Difference ± SD for Before-After Comparison**			**Mean Difference (CI) for Between-Group Comparison**				
					**Pre-Post**		**Pre-Follow-Up**	***p* value**	**Post Intervention**	***p* value**	**Follow-Up**	***p* value**
Intervention *n =* 59	4.21 ± 2.33	5.06 ± 2.13	4.92 ± 2.23		0.98 ± 1.90		0.76 ± 1.92	Not significant *	0.38 (−0.24 to 1.00)	Not significant	−0.02 (−0.66 to 0.62)	Not significant
Control *n* = 61	3.66 ± 1.75	4.43 ± 2.23	4.76 ± 2.07		0.66 ± 1.95		0.97 ± 1.84	Not significant *				
**DLCO%:** Diffusing lung capacity for carbon monoxide is a measurement to assess the lungs’ ability to transfer gas from inspired air to the bloodstream.												
**Liu et al., 2020**	**Pre**		**Post**		**Mean Difference for Before-After Comparison**			**Mean Difference (CI) for Between-Group Comparison**				
					***p* value**			**Post Test**			***p* value**	
Intervention *n* = 36	60.3 ± 11.3		78.1 ± 12.3		*p* < 0.05			4.98 (9.05 to 21.15) *			*p* < 0.05	
Control *n* = 36	60.7 ± 12.0		63.0 ± 13.4		Not significant							
**Muscle Mass and Strength**												
**Muscle Mass:** Muscle mass, measured by means of a magnetic resonance imaging (MRI) scan.												
**MRI—mid arm: cm^2^**												
**Nambi et al., 2022**	**Pre**	**Post**	**Follow-Up 1**	**Follow-Up 2**	**Mean Difference for Before-After Comparison**			**Mean Difference for Between-Group Comparison**				
					**Pre-Post**	**Pre-Follow-Up 1**	**Pre-Follow-Up 2**	**Post Intervention**		**Follow-Up 1**		**Follow-Up 2**
Intervention *n* = 38	56.3 ± 1.1	57.9 ± 0.9	59.0 ± 0.5	61.5 ± 0.2	*p* < 0.001	*p* < 0.001	*p* < 0.001	1.88 (−0.03 to 0.83) * Not significant *		0.79 (−0.15 to 0.35) * Not significant *		0.15 (−0.07 to 0.27) * Not significant *
Control *n* = 38	55.9 ± 1.7	57.5 ± 1.0	58.9 ± 0.6	61.4 ± 0.5	*p* < 0.001	*p* < 0.001	*p* < 0.001					
**MRI—mid thigh: cm^2^**												
**Nambi et al., 2022**	**Pre**	**Post**	**Follow-Up 1**	**Follow-Up 2**	**Mean Difference for Before-After Comparison**			**Mean Difference for Between-Group Comparison**				
					**Pre-Post**	**Pre-Follow-Up 1**	**Pre-Follow-Up 2**	**Post Intervention**		**Follow-Up 1**		**Follow-Up 2**
Intervention *n* = 38	63.5 ± 0.8	65.5 ± 0.6	68.5 ± 0.6	72.6 ± 0.8	*p* < 0.001	*p* < 0.001	*p* < 0.001	0.45 (−0.07 to 0.47) * Not significant *		0.73 (−0.17 to 0.37) * Not significant *		0.55 (−0.27 to 0.47) * Not significant
Control *n* = 38	63.4 ± 0.8	65.3 ± 0.6	68.4 ± 0.6	72.5 ± 0.8	*p* < 0.001	*p* < 0.001	*p* < 0.001					
**MRI—mid calf: cm^2^**												
**Nambi et al., 2022**	**Pre**	**Post**	**Follow-Up 1**	**Follow-Up 2**	**Mean Difference for Before-After Comparison**			**Mean Difference for Between-Group Comparison**				
					**Pre-Post**	**Pre-Follow-Up 1**	**Pre-Follow-Up 2**	**Post Intervention**		**Follow-Up 1**		**Follow-Up 2**
Intervention *n* = 38	60.2 ± 1.1	65.2 ± 0.6	66.4 ± 0.5	68.7 ± 0.5	*p* < 0.001	*p* < 0.001	*p* < 0.001	0.00 (−0.27 to 0.27) * Not significant *		0.87 (−0.13 to 0.33) * Not significant *		0.00 (−0.23 to 0.23) * Not significant *
Control *n* = 38	60.2 ± 1.1	65.2 ± 0.6	66.3 ± 0.5	68.7 ± 0.5	*p* < 0.001	*p* < 0.001	*p* < 0.001					
**Static Squat Test at the wall:** A strength measure that assesses the time, in seconds, that participants can remain in a squatting position against a wall with both feet flat on the ground, approximating a 90° angle at the hip and knees.												
**Li et al., 2021**	**Pre**	**Post**	**Follow-Up**		**Mean Difference ± SD for Before-After Comparison**			**Mean Difference (CI) for Between-Group Comparison**				
					**Pre-Post**	**Pre-Follow-Up**	***p* value**	**Post Intervention**		***p* value**	**Follow-Up**	***p* value**
Intervention *n =* 59	34.68 ± 21.85	63.67 ± 37.33	61.46 ± 36.33		29.35 ± 27.22	28.12 ± 27.17	<0.001 *	20.12 (12.34 to 27.90)		<0.001	22.23 (14.24 to 30.21)	<0.001
Control *n* = 61	38.60 ± 25.07	46.58 ± 30.55	41.56 ± 24.30		7.98 ± 19.53	4.16 ± 19.62	<0.001 *					
**Handgrip strength:** Test used to measure upper limb strength using a handheld dynamometer in Kg.												
**Nambi et al., 2022**	**Pre**	**Post**	**Follow-Up 1**	**Follow-Up 2**	**Mean Difference for Before-After Comparison**			**Mean Difference for Between-Group Comparison**				
					**Pre-Post**	**Pre-Follow-Up 1**	**Pre-Follow-Up 2**	**Post Intervention**			**Follow-Up 1**	**Follow-Up 2**
Intervention *n* = 38	28.5 ± 0.6	29.2 ± 0.6	29.8 ± 0.5	30.4 ± 0.8	*p* < 0.001	*p* < 0.001	*p* < 0.001	−1.58 (−0.45 to 0.05) * Not significant *			−13.42 (−1.95 to −1.45) * *p* value < 0.001 *	−21.25 (−4.27 to −3.53) * *p* value 0.003 *
Control *n* = 38	28.4 ± 0.7	29.4 ± 0.5	31.5 ± 0.6	34.3 ± 0.8	*p* < 0.001	*p* < 0.001	*p* < 0.001					
**Functional Exercise Capacity**												
**6MWT:** The 6-min walk test assesses distance walked (in meters) over 6 min as a sub-maximal test of aerobic capacity/endurance. Assistive devices can be used.												
**Li et al., 2021**	**Pre**	**Post**	**Follow Up**		**Mean Difference ± SD for Before-After Comparison**			**Mean Difference (CI) for Between-Group Comparison**				
					**Pre-Post**	**Pre-Follow-Up**	***p* value**	**Post Intervention**		***p* value**	**Follow-Up**	***p* value**
Intervention *n =* 59	514.52 ± 82.87	588.40 ± 63.39	590.58 ± 69.67		80.20 ± 74.66	84.81 ± 80.38	<0.001 *	65.45 (43.80 to 87.10)		<0.001	68.62 (46.39 to 90.85)	<0.001
Control *n* = 61	499.98 ± 93.41	517.07 ± 88.87	521.38 ± 93.11		17.09 ± 63.94	15.17 ± 70.02	<0.001 *					
**Liu et al., 2020**	**Pre**		**Post**		**Mean Difference for Before-After Comparison**			**Mean Difference (CI) for Between-Group Com-parison**				
					***p* value**			**Post Test**			***p* value**	
Intervention *n* = 36	162.7 ± 72.0		212.3 ± 82.5		<0.05			3.03 (18.77 to 91.43) *			*p* < 0.05	
Control *n =* 36	155.7 ± 82.1		157.2 ± 71.7		Not significant							
**STS:** The 30-Second Sit to Stand Test is used to test leg strength and endurance. The participant is encouraged to complete as many full stands as possible within 30 s.												
**De Souza et al., 2021**	**Pre**		**Post**		**Mean Difference for Before-After Comparison**			**Mean Difference (CI) for Between-Group Comparison**				
					***p* value**			**Post**			***p* value**	
Intervention *n* = 104	12.7 ± 3.2		19.5 ± 3.1		<0.05			5.4 (3.6 to 9.1)			<0.001	
Control *n* = 92	13.1 ± 2.9		14.5 ± 3.3		Not significant							
**PADL:** Physical Activity in Daily Life is assessed using a mobile phone app to measure the steps taken in a day.												
**De Souza et al., 2021**	**Pre**		**Post**		**Mean Difference for Before-After Comparison**			**Mean Difference (CI) for Between-Group Comparison**				
					***p* value**			**Post**			***p* value**	
Intervention *n* = 104	8671 ± 1355		10492 ± 1122		<0.05			1716 (975 to 2335)			<0.001	
Control *n* = 92	8958 ± 1744		9063 ± 1201		Not significant							

Data were extracted by the original reports, or through correspondence with the authors, or they were calculated based on available data (*), if possible, when it was not possible to obtain them from the original authors. In these results, *n* refers to the number of participants included in the analyses and is not necessarily equivalent to the number enrolled at the baseline or retained at the follow-up. *CI*, confidence interval.

**Table 4 ijerph-19-05185-t004:** Quality of life and independence outcomes.

QUALITY OF LIFE AND INDEPENDENCE OUTCOMES													
**ADL and QoL**													
**SF-12 PCS:** 12-Short Forum Health Survey, Physical Component Summary.													
**Li et al** **., 2021**	**Pre**		**Post**		**Follow** **-Up**		**Mean Difference ± SD for Before-After Comparison**			**Mean Difference (CI) for Between-Group Comparison**			
							**Pre-Post**	**Pre-Follow** **-Up**	***p* value**	**Post Intervention**	***p* value**	**Follow** **-Up**	***p* value**
Intervention *n =* 59	39.15 ± 7.16		47.13 ± 6.83		47.38 ± 8.22		7.81 ± 7.02	8.2 ± 10.05	0.003 *	3.79 (1.24 to 6.35)	0.004	2.69 (0.06 to 5.32)	Not significant
Control *n* = 61	39.69 ± 7.06		43.53 ± 8.12		45.10 ± 8.23		3.84 ± 7.60	5.20 ± 9.13	Not significant *				
**SF-12 MCS:** 12-Short Forum Health Survey, Mental Component Summary.													
**Li et al** **., 2021**	**Pre**		**Post**		**Follow** **-Up**		**Mean Difference ± SD for Before-After Comparison**			**Mean Difference (CI) for Between-Group Comparison**			
							**Pre-Post**	**Pre-Follow** **-Up**	***p* value**	**Post Intervention**	***p* value**	**Follow** **-Up**	***p* value**
Intervention *n =* 59	44.67 ± 8.76		50.80 ± 7.83		52.08 ± 7.26		6.15 ± 10.78	6.92 ± 10.28	Not significant *	2.18 (−0.54 to 4.90)	Not significant	1.99 (−0.81 to 4.79)	Not significant
Control *n* = 61	44.13 ± 8.25		48.30 ± 8.65		49.61 ± 8.68		4.17 ± 8.79	5.51 ± 7.79	Not significant *				
**SarQoL:** Sarcopenia and Quality of Life. The SarQoL questionnaire is designed to assess the quality of life of sarcopenic patients.													
**Nambi et al** **., 2022**	**Pre**	**Post**		**Follow** **-Up 1**		**Follow** **-Up 2**	**Mean Difference for Before-After Comparison**			**Mean Difference for Between-Group Comparison**			
							**Pre-Post**	**Pre-Follow** **-Up 1**	**Pre-Follow** **-Up 2**	**Post Intervention**	**Follow** **-Up 1**		**Follow** **-Up 2**
Intervention *n* = 38	57.7 ± 1.0	58.8 ± 0.9		60.5 ± 0.8		62.2 ± 0.8	*p* < 0.001	*p* < 0.001	*p* < 0.001	−22.71 (−4.57 to −3.83) * *p* value <0.001 *	−40.92 (−8.91 to −8.09) * *p* value <0.001 *		−50.06 (−10.81 to −9.99) * *p* value <0.001 *
Control *n =* 38	57.3 ± 1.0	63.0 ± 0.7		69.0 ± 1.0		72.6 ± 1.0	*p* < 0.001	*p* < 0.001	*p* < 0.001				
**SF-36:** 36-Short Form Health Survey													
**Liu et al., 2020**	**Pre**			**Post**			**Mean Difference for Before-After Comparison**			**Mean Difference (CI) for Between-Group Comparison**			
							** *p* ** **value**			**Post Test**		** *p* ** **value**	
** *Physical health* **													
Intervention *n = 36*	52.4 ± 6.2			71.6 ± 7.6			*p* < 0.05			9.83 (13.95 to 21.05) *		*p* < 0.05	
Control *n* = 36	53.2 ± 7.7			54.1 ± 7.5			*p* < 0.05						
** *Body role function* **													
Intervention *n* = 36	61.2 ± 6.6			75.9 ± 7.9			*p* < 0.05			7.75 (10.32 to 17.48) *		*p* < 0.05	
Control *n* = 36	61.3 ± 7.2			62.0 ± 7.3			*p* < 0.05						
** *Physical pain* **													
Intervention *n* = 36	63.5 ± 7.4			78.3 ± 7.8			*p* < 0.05			8.32 (11.71 to 19.09) *		*p* < 0.05	
Control *n* = 36	63.5 ± 8.1			62.9 ± 7.9			*p* < 0.05						
** *General health* **													
Intervention *n* = 36	61.8 ± 7.7			74.2 ± 7.9			*p* < 0.05			7.32 (9.31 to 16.29) *		*p* < 0.05	
Control *n* = 36	61.8 ± 8.4			61.4 ± 6.9			*p* < 0.05						
** *Energy* **													
Intervention *n* = 36	60.6 ± 6.9			75.6 ± 7.1			*p* < 0.05			9.10 (11.24 to 17.56) *		*p* < 0.05	
Control *n* = 36	60.5 ± 7.1			61.2 ± 6.3			*p* < 0.05						
** *Social function* **													
Intervention *n* = 36	59.4 ± 7.2			69.8 ± 6.4			*p* < 0.05			7.11 (7.84 to 13.96) *		*p* < 0.05	
Control *n* = 36	59.5 ± 7.0			58.9 ± 6.6			*p* < 0.05						
** *Emotional role function* **													
Intervention *n* = 36	61.4 ± 6.9			75.7 ± 7.0			*p* < 0.05			8.84 (11.54 to 18.26) *		*p* < 0.05	
Control *n* = 36	61.4 ± 7.3			60.8 ± 7.3			*p* < 0.05						
** *Mental health* **													
Intervention *n* = 36	61.5 ± 6.5			73.7 ± 7.6			*p* < 0.05			6.48 (8.03 to 15.17) *		*p* < 0.05	
Control *n* = 36	61.6 ± 7.2			62.1 ± 7.6			*p* < 0.05						
**Independence Outcomes**													
**FIM:** Functional Independence Measure. The FIM is used to assess and grade the functional status of a person based on the level of assistance required.													
**Liu et al., 2020**	**Pre**			**Post**			**Mean Difference for Before-After Comparison**			**Mean Difference (CI) for Between-Group Comparison**			
							** *p* ** **value**			**Post Test**		** *p* ** **value**	
Intervention *n* = 36	109.2 ± 13			109.4 ± 11.1			Not significant			0.2 (−4.47 to 5.47) *		Not significant	
Control *n* = 36	109.3 ± 10.7			108.9 ± 10.1			Not significant						

Data were extracted by the original reports or through correspondence with the authors, or they were calculated based on available data (*), if possible, when it was not possible to obtain them from the original authors. In these results, *n* refers to the number of participants included in the analyses and is not necessarily equivalent to the number enrolled at the baseline or retained at the follow-up. *CI*, confidence interval.

## Data Availability

Data is contained within this manuscript or in the Appendix A.

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
