# Peer review of "Rehabilitation Interventions for Post-Acute COVID-19 Syndrome: A Systematic Review"

_ijerph, 2022, doi:10.3390/ijerph19095185_

Round 1

Reviewer 1 Report

This is an interesting report of a systematic review of the effects of COVID-19 on post-acute rehabilitation. The conclusions of this review are of social and global importance and should be studied and applied in daily life by readers (scientific and non-scientific fields). The conclusions validated in the systematic review were not a leap of faith, but an appropriate interpretation.

Author Response

Thank you. We are honored by your appreciation of our work.

Reviewer 2 Report

Thank you for allowing me to read this paper which initially deals with an interesting and relevant topic. The suggestions given in this document are intended to improve your work.

First, authors are advised to review the journal's instructions: https://www.mdpi.com/journal/ijerph/instructions

General and major comments:

  • Many of the formats provided in the word template are not respected; even some tables are not readable.
  • A review of writing and English is recommended.
  • Authors should check the back matter information.
  • Are the authors sure they have created the reference list as recommended by the journal?
  • The authors refer to a series of numbered appendices not included in the work. I would like to read them, please. Also, the appendices should be named with letters.
  • Please place figures and tables immediately after they are mentioned.
  • Please provide the PRISMA checklist as additional material. http://prisma-statement.org/documents/PRISMA_2020_checklist.pdf

Introduction section:

  • COVID needs to be better contextualised. I think the authors should better explain the conditions covered by PACS: 1) long COVID and 2) sequela by COVID. More weight should be given to the great variability of symptomatology and how it influences the subjects' day-to-day life and quality of life.
  • This section must end with the aim/s. Although two general objectives are provided, later, there is a section called objectives with another. It is confusing. Isn't the research question too open-ended for a systematic review? Do you consider it to be a systematic review or perhaps a scoping review?

Methods section:

  • This section could be better structured, and information needs to be included or expanded. I recommend that the authors read other systematic reviews published in the journal and the PRISMA checklist to better understand this section's structure.
  • Please be clearer on the eligibility criteria.
  • How was data extraction done, and with which tools? How were discrepancies resolved?

Results section:

  • There is a lot of information, and it is disorganised. Please try to be clearer and more concise.

Discussion and conclusion sections:

  • The discussion should be better organised and include: the main findings, the theoretical and practical implications of this study, limitations, and future directions. Although some information is provided, the writing of the main findings, ordered by type of symptomatology, type of intervention or the rest of the parameters studied by the authors, comparing their results with those of other studies, should be improved.
  • Conclusions should only refer to your findings.

Reviewer 3 Report

The paper presents a review of five randomized controlled trials (RCTs) in order to investigate the effectiveness of different rehabilitation interventions in the post-acute phase of COVID-19. This is a useful and detailed systematic review.

Minor corrections:

1. In the section Search strategy, Appendix 1 is mentioned. However, at the end of the paper Appendix A is listed. Whether it should be renamed? Also, figures and tables do not have the same label in the text and in the appendix.

2. Figure 2: there is no “minus” in the table, so “High risk of bias” could be removed from the bottom of the table.

3. Page 14, Table 4.a: for SAS and SDS there is a sentence that starts with “Is a method…” It could be reformulated to the affirmative sentence in the third person singular.

Round 2

Reviewer 2 Report

I think the authors have made a great effort to improve the paper.